# Which urban land covers/uses are associated with residents' mortality? A cross-sectional, ecological, pan-European study of 233 cities

Jonathan R Olsen ![ORCID],[1] Natalie Nicholls,[1] Graham Moon,[2] Jamie Pearce,[3] Niamh Shortt,[3] Richard Mitchell[1]

[1]MRC/CSO Social and Public Health Sciences Unit, University of Glasgow, Glasgow, UK
[2]School of Geography and Environmental Sciences, University of Southampton, Southampton, UK
[3]School of Geosciences, University of Edinburgh, Edinburgh, UK

**Correspondence to**
Dr Jonathan R Olsen;
jonathan.olsen@glasgow.ac.uk

## ABSTRACT

**Objectives** The study aim was to determine whether the range and distribution of all, and proportions of specific, land covers/uses within European cities are associated with city-specific mortality rates.

**Setting** 233 European cities within 24 countries.

**Participants** Aggregated city-level all-cause and age-group standardised mortality ratio for males and females separately and Western or Eastern European Region.

**Results** The proportion of specific land covers/uses within a city was related to mortality, displaying differences by macroregion and sex. The land covers/uses associated with lower standardised mortality ratio (SMR) for both Western and Eastern European cities were those characterised by 'natural' green space, such as forests and semi-natural areas (Western Female coefficient: −18.3, 95% CI −29.8 to −6.9). Dense housing was related to a higher SMR, most prominently in Western European cities (Western Female coefficient: 17.4, 95% CI 9.6 to 25.2).

**Conclusions** There is pressure to build on urban natural spaces, both for economic gain and because compact cities are regarded as more sustainable, yet here we offer evidence that doing so may detract from residents' health. Our study suggests that urban planners and developers need to regard retaining more wild and unstructured green space as important for healthy city systems.

## BACKGROUND

In 2014, 54% of the world's population lived in urban environments; this is set to rise to 66% by 2050.[1] It is therefore important that urban environments are designed to enable and promote good health. Urban areas are complex systems and one emergent feature is the health of their residents.[2] There are stark between-city inequalities in population health[3] and previous work on trends and inequalities between European cities in sex-specific mortality rates suggests that a city environment itself is an important influence.[4–6]

The physical extent, design and composition of cities are important components of the urban system and are associated with the health and well-being of residents.[7–9] The

### Strengths and limitations of this study

► This was a large international study of 233 cities within 24 countries.
► We used high-quality, internationally comparable land cover/use data from the European Urban Atlas.
► We were able to examine the impact of land covers/uses both collectively (via landscape metrics) and individually.
► Previously validated city-level standardised mortality ratio (SMR) data were spatially joined with contemporaneous land cover/use data.
► Although associations between land covers/uses and SMRs were described, we have no information on how the individuals used or were exposed to the land covers/uses.

diverse development trajectories of cities means there is considerable variety in their size, configuration and content.[10] This variety provides a rich natural laboratory to study the implications for health of their design and construction. Given that most cities are subject to some kind of planning control and vision, there is also great potential for epidemiological insights into how the urban landscape affects health to impact on the lived experiences of billions of people.

This study is about 'what is on the ground' in a city, which we refer to as land cover/uses. Land cover refers to what the surface actually is, for example, forests, concrete or water. Land uses are the purposes for which humans use the land, such as airports or roads. Urban land cover/use can influence health via multiple mechanisms including the promotion or inhibiting of healthy behaviours,[11] increasing exposure to pathogens such as air pollutants and noise[12] or offering psychologically restorative spaces.[13]

Evidence suggests that land cover/use exerts an independent influence on

residents, over and above residents' own characteristics. For example, the density of settlement and transport networks have been associated with variation in mortality risk across a number of English cities.[14] In a previous multicity European study, we found that the presence and quantity of specific land covers/uses within a city was associated with residents' reported life satisfaction. We also found that specific land covers/uses, and a more even distribution of the land covers/uses (as opposed to dominance by one or two land covers/uses), were both associated with lower levels of socioeconomic inequality in life satisfaction within the city.[15] We hypothesised that the range and distribution of land covers/uses speaks to the range of opportunities and environments, and hence affordances[16] that the city provides—and that this might explain the associations with life satisfaction levels and its equality. However, life satisfaction is fundamentally a subjective measure of well-being, and may be particularly sensitive to the environment. Therefore, in this study we focused on mortality which is, arguably, a more robust measure of population health. In doing so, we were able to include a substantially larger sample of European cities (n=233). No previous study has assessed associations between city-level land covers/uses and mortality in such a large and diverse international sample of cities.

The study aim was to determine whether the range and distribution of all, and proportions of specific, land covers/uses within European cities is associated with the city-specific mortality rate. Previous research found that a city's wider macroregion (ie, its location in Eastern or Western Europe) was an important determinant of mortality rate, and moderator of association between environment and health,[5 17] therefore this was also explored. Our specific research questions were:

i. Is the range and distribution of land covers/uses within a city associated with city-level mortality rates?
ii. Which, if any, specific land covers/uses are associated with city-level mortality rates?
iii. How do these between associations vary with sex and by macroregion?

## METHODS
### Setting and data sources
#### European Urban Atlas
The 2006 Urban Atlas provided pan-European comparable land cover/use data collected and digitised from satellite data for urban areas.[18] We selected data for this period to match the time period for the available mortality data. The Atlas distinguishes 26 different land cover/use categories at a $10\,m^2$ resolution.[19] Figure 1 provides an illustration of these data for part of one city. In this example, 20 different land uses are shown and labelled. Full land use descriptions are available in online supplementary table 1. The land cover/use data were matched to the same city administrative boundaries as the mortality data.

#### Geoprocessing of Urban Atlas data
We calculated the total area ($km^2$) for each land cover/use by city, using an extension for ArcMap GIS called Patch Analyst.[20] This was then converted to proportion of land cover/use by city (some values equalled zero if the land cover/use was not present in a city, for example, not all cities contained an airport). Following our previous work,[15] we then calculated landscape metrics. These are quantitative summaries of the range and statistical distribution of the various land covers/uses within a whole city. We calculated Shannon's diversity index (SDI) and Shannon's Evenness index (SEI)[21] to assess the variety of all land covers/uses, and their relative availability, within each city. Only land covers/uses present within each city were included, this provided a relative measure of the diversity of land covers/uses available at the city level (SDI) and the distribution of area among the land covers/uses present within a city (SEI).

#### Gross domestic product
Following Richardson et al,[3] the gross domestic product (GDP) per 1000 capita for the city region was used as a measure of affluence/poverty for that city. Historic GDP per country for the years 2003–2006 inclusive were extracted from Eurostat and averaged to provide a single measure for each city.

#### European mortality data (standardised mortality ratios)
Mortality was measured using all-cause and age-group standardised mortality ratios (SMR) for males and females separately, calculated for 233 European cities by Richardson et al.[3] Values >100 indicated a higher level of mortality than expected given the time period and age structure of the city population, and values <100 indicated a lower level. The SMR data by Richardson et al covered the time period 1999–2009, divided into three waves; the second of which was 2003–2006. This time period most closely matched the date of the land cover measures from the Urban Atlas. The mortality indicator was at the city-level. No information was available about mortality rates either for socioeconomic groups or spatial locations within the city. We could not, therefore, assess within-city inequalities in this study.

The SMR, land cover/use and GDP data were matched at city level. The final dataset included 233 cities (within 24 countries). The cities were allocated to either Western or Eastern-Central bloc macroregion, following Richardson et al,[3] referred to hereafter as Western or Eastern. There were 85 cities (in 11 countries) classed as Eastern.

#### Patient and public involvement
No patients were involved in the development of the research question, design and implementation of the study or interpretation of the results.

#### Statistical analysis
First, linear regression models assessed associations between the landscape metrics and city SMR data, adjusting for city GDP. Next, models for specific land

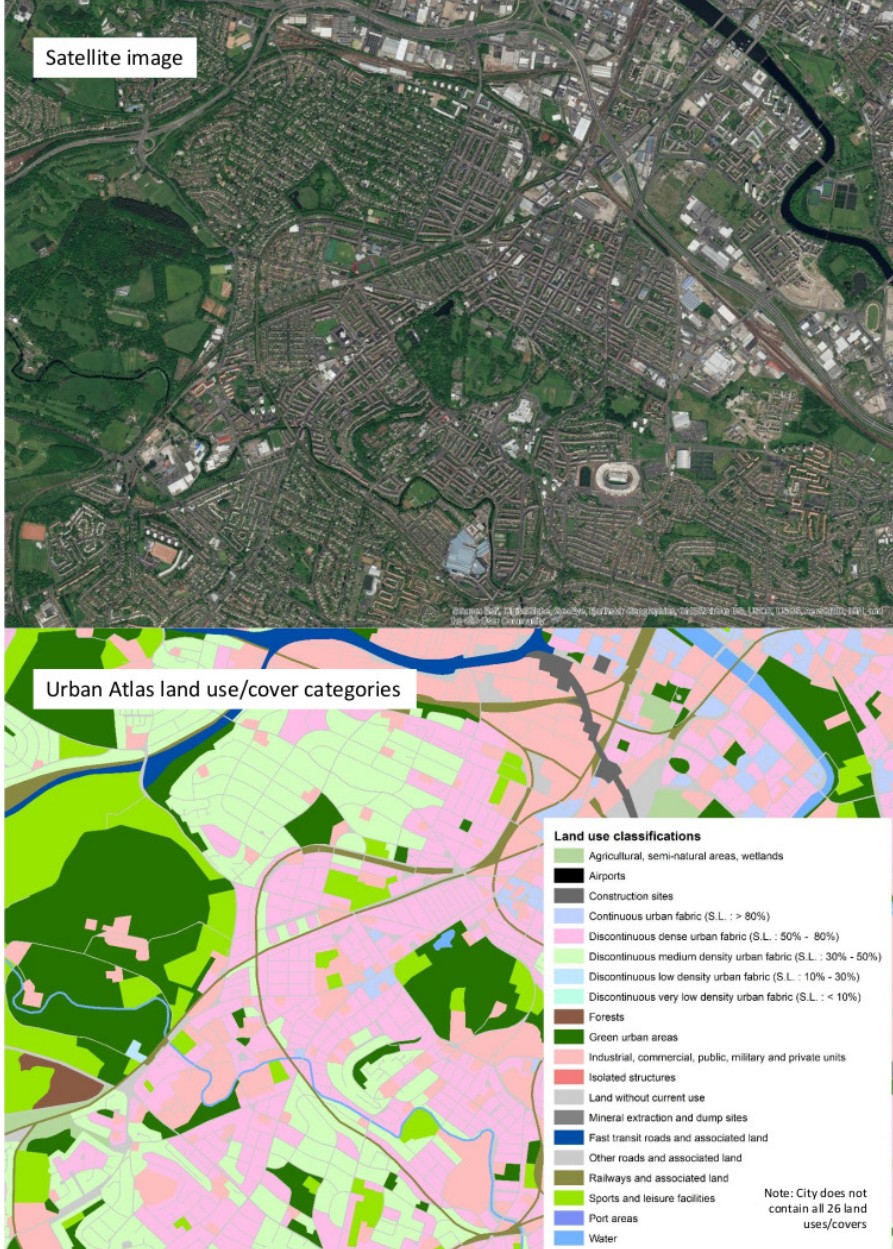

**Figure 1** Urban Atlas land use/cover categories and satellite base map.

covers/uses were built. Preliminary analyses suggested that associations between the specific land cover/use measures and SMR were not linear, so the former were categorised. To do this, the proportion of the total area covered by each land measure was calculated within each city, and then classified into quintiles. This process was carried out separately for each macroregion. We categorised the data into quintiles rather than apply polynomial terms in order to aid interpretation and therefore help guide policymakers.[22] Furthermore, this approach also allowed us to generate the marginal estimates, which were plotted. These quintiles were then fit as factors, along with city GDP, in a linear regression, while also allowing for clustering within country. Each land measure was assessed separately using Wald statistics and subjected to post hoc pairwise comparisons using Sidak's correction

for multiple testing across all terms.[23] We chose not to include all land covers/uses in a singular model due to concern of multicollinearity, which was confirmed when tested (Eastern European Cities—VIF: 17.20, Western European Cities—VIF: 6.41).[24 25] All analyses were run separately by sex and macroregion. Stata (SE V.14)[26] and R (V.3.6.0)[27] were used for analysis, with significance levels set at 5%.

## RESULTS

For the models looking at specific land covers/uses we present our results in three ways. First, we highlight the land covers/uses for which we found strongest evidence of association with either lower or higher mortality rates (table 1). This selection was systematic, based on Wald

**Table 1** Land covers/uses with strongest and most consistent evidence of association with city SMRs

| Universal (both regions and sexes) | | | |
|---|---|---|---|
| **West (M&F)** | ***Trend*** | **East (M&F)** | ***Trend*** |
| Agricultural, semi-natural areas, wetlands | ↓ SMR | Agricultural, semi-natural areas, wetlands | ↓ SMR |
| **Partially universal (across either both regions and/or sexes)** | | | |
| <u>West F only</u> | ***Trend*** | **East (M&F)** | ***Trend*** |
| Green urban areas | ↑ SMR | Green urban areas | ↑ SMR |
| **West (M&F)** | ***Trend*** | **East F only** | ***Trend*** |
| Industrial, commercial, public, military | ↑ SMR | Industrial, commercial, public, military | ↑ SMR |
| **Regional (both sexes)** | | | |
| **West (M&F)** | ***Trend*** | **East (M&F)** | ***Trend*** |
| Discontinuous low density urban fabric* | ↑ SMR | | |
| Residential | ↑ SMR | | |
| | | Sports and leisure facilities | ↓ SMR |
| | | Land without current use | ↑ SMR |
| **Sex (both regions)** | | | |
| **Males** | ***Trend*** | **Females (E&W)** | ***Trend*** |
| | | Forests | ↓ SMR |
| **Regions and opposite sex** | | | |
| **West M** | ***Trend*** | **East F** | ***Trend*** |
| Isolated structures | ↓ SMR | Isolated structures | ↓ SMR |

*Most evident in cities with the highest proportion of this land cover/use (quintile 5).
SMR, standardised mortality ratio.

test values and on both size and apparent 'dose response' of associations between quintiles of land cover/use and mortality. Table 1 crudely 'ranks' these selected land covers/uses according to the extent to which they were associated with mortality in both macroregions, and both sexes (which we call 'universal'). A downward pointing arrow denotes that more of that land cover/use was associated with lower mortality, and an upward arrow that it was associated with higher mortality. Second, we present regression coefficients for these selected land covers/uses (table 2). Online supplementary table 2 (Western Europe) and online supplementary table 3 (Eastern Europe) provide the full set of regression coefficients for *all* land covers/uses, together with Wald statistics. Third, we graph marginal means from the adjusted regression models for associations identified in table 1 as being either *universal* or *partially universal* (figures 2 and 3). These figures convert the associations to estimated SMRs, perhaps a more meaningful expression of the results. The reader is invited to pay attention to the position on the Y axis, and slope, of the lines in the figures. We found no meaningful associations between SMRs and the landscape metrics and therefore do not report those results.

### Land covers/uses associated with lower mortality

In general, cities with a higher proportion of *agricultural, semi-natural areas and wetlands* enjoyed lower mortality rates (figure 2). In Eastern cities, this advantage weakened in cities with the *highest* proportion of this land use (figure 2, table 2). Figure 2 shows a somewhat large difference in SMR between both sexes for Western cities in quintile 1 and quintile 5. A higher proportion of *forests* within a city was also associated with lower SMR for females in both Eastern and Western European cities, but not males (table 2). In Western cities, the proportion of *isolated structures* (*single dwellings surrounded on all sides with natural green land*) was associated with lower SMRs for males, but in Eastern cities this protective association was present for females only (table 2). For Eastern European cities only, higher proportion of land containing *sports and leisure facilities* was associated with lower SMRs, although only significantly for men (table 2).

### Land covers/uses associated with higher mortality

The two land covers/uses most consistently associated with higher mortality were *industrial, commercial, public and military*, and *green urban areas* (figure 3). In Western European cities, higher proportions of the former were associated with higher SMRs, with a sharpening of association

**Table 2** Regression coefficients of land covers/uses found to contribute to a model for a particular macroregion/sex based on Wald test measure (significant results marked (<0.05) in bold font)

| Quintiles (proportion of land use) | West Males | | | | West Females | | | | East Males | | | | East Females | | | |
|---|---|---|---|---|---|---|---|---|---|---|---|---|---|---|---|---|
| | Coefficient | LL 95% CI | UL 95% CI | P value | Coefficient | LL 95% CI | UL 95% CI | P value | Coefficient | LL 95% CI | UL 95% CI | P value | Coefficient | LL 95% CI | UL 95% CI | P value |
| **Agricultural, semi-natural areas, wetlands** | | | | | | | | | | | | | | | | |
| 1 | Reference | | | | Reference | | | | Reference | | | | Reference | | | |
| 2 | −10.22 | −17.69 | −2.75 | 0.011 | **−11.66** | **−20.75** | **−2.58** | **0.016** | **−14.45** | **−30.34** | **1.44** | **0.07** | −3.68 | −12.24 | 4.88 | 0.36 |
| 3 | **−11.67** | **−17.30** | **−6.04** | **0.001** | **−13.03** | **−23.12** | **−2.94** | **0.015** | −11.51 | −30.34 | 7.32 | 0.203 | −2.86 | −10.81 | 5.08 | 0.441 |
| 4 | **−12.63** | **−19.14** | **−6.13** | **0.001** | **−19.71** | **−33.78** | **−5.65** | **0.01** | **−18.67** | **−37.03** | **−0.32** | **0.047** | −7.14 | −16.67 | 2.39 | 0.126 |
| 5 | **−13.66** | **−21.08** | **−6.23** | **0.002** | **−18.32** | **−29.77** | **−6.88** | **0.004** | −3.89 | −26.56 | 18.78 | 0.71 | 5.76 | −8.51 | 20.03 | 0.39 |
| | *(Wald test of parameter: 0.006)* | | | | *(Wald test of parameter: 0.051)* | | | | *(Wald test of parameter: 0.017)* | | | | *(Wald test of parameter: 0.014)* | | | |
| **Forests** | | | | | | | | | | | | | | | | |
| 1 | Reference | | | | Reference | | | | Reference | | | | Reference | | | |
| 2 | −4.60 | −10.36 | 1.16 | 0.11 | **−9.71** | **−18.44** | **−0.98** | **0.03** | −6.93 | −20.41 | 6.55 | 0.28 | **−13.55** | **−24.82** | **−2.28** | **0.02** |
| 3 | −4.53 | −10.66 | 1.60 | 0.13 | −8.48 | −18.81 | 1.85 | 0.10 | −5.82 | −13.75 | 2.11 | 0.13 | **−10.43** | **−16.25** | **−4.62** | **0.00** |
| 4 | −9.54 | −19.44 | 0.37 | 0.06 | −15.13 | −31.88 | 1.62 | 0.07 | −0.73 | −12.67 | 11.21 | 0.89 | −8.27 | −19.36 | 2.83 | 0.13 |
| 5 | **−9.03** | **−16.99** | **−1.08** | **0.03** | **−12.59** | **−25.08** | **−0.11** | **0.05** | −4.99 | −13.08 | 3.10 | 0.20 | **−15.96** | **−24.74** | **−7.17** | **0.00** |
| | *(Wald test of parameter: 0.156)* | | | | *(Wald test of parameter: 0.005)* | | | | *(Wald test of parameter: 0.309)* | | | | *(Wald test of parameter: 0.010)* | | | |
| **Green urban areas** | | | | | | | | | | | | | | | | |
| 1 | Reference | | | | Reference | | | | Reference | | | | Reference | | | |
| 2 | 1.60 | −4.47 | 7.66 | 0.58 | −0.02 | −7.11 | 7.07 | 1.00 | 1.48 | −6.00 | 8.95 | 0.67 | 4.07 | −8.47 | 16.62 | 0.49 |
| 3 | 3.96 | −2.05 | 9.96 | 0.18 | 1.43 | −7.81 | 10.67 | 0.74 | **−5.99** | **−12.37** | **0.39** | **0.06** | −3.06 | −12.89 | 6.77 | 0.50 |
| 4 | **7.23** | **−0.26** | **14.73** | **0.06** | **9.57** | **1.21** | **17.93** | **0.03** | **−10.07** | **−19.96** | **−0.19** | **0.05** | −9.26 | −23.29 | 4.77 | 0.17 |
| 5 | **11.33** | **2.44** | **20.21** | **0.02** | **15.06** | **4.45** | **25.67** | **0.01** | **15.37** | **3.66** | **27.08** | **0.02** | 3.85 | −8.48 | 16.19 | 0.50 |
| | *(Wald test of parameter: 0.132)* | | | | *(Wald test of parameter: 0.029)* | | | | *(Wald test of parameter: 0.001)* | | | | *(Wald test of parameter: 0.004)* | | | |
| **Industrial, commercial, public, military** | | | | | | | | | | | | | | | | |
| 1 | Reference | | | | Reference | | | | Reference | | | | Reference | | | |
| 2 | 2.02 | −2.51 | 6.55 | 0.35 | −0.23 | −6.54 | 6.08 | 0.94 | −12.53 | −29.98 | 4.91 | 0.14 | −0.92 | −18.26 | 16.43 | 0.91 |
| 3 | 1.34 | −5.24 | 7.93 | 0.67 | 1.75 | −6.93 | 10.42 | 0.67 | −1.70 | −16.73 | 13.32 | 0.81 | 4.72 | −9.75 | 19.20 | 0.48 |
| 4 | 3.24 | −2.16 | 8.65 | 0.22 | **6.76** | **2.31** | **11.21** | **0.01** | −7.70 | −20.69 | 5.30 | 0.22 | −1.34 | −12.77 | 10.08 | 0.80 |
| 5 | **10.64** | **3.79** | **17.50** | **0.01** | **11.02** | **0.93** | **21.11** | **0.04** | −2.81 | −11.94 | 6.31 | 0.51 | 6.09 | −0.19 | 12.37 | 0.06 |
| | *(Wald test of parameter: 0.049)* | | | | *(Wald test of parameter: 0.023)* | | | | *(Wald test of parameter: 0.491)* | | | | *(Wald test of parameter: 0.033)* | | | |
| **Isolated structures** | | | | | | | | | | | | | | | | |
| 1 | Reference | | | | Reference | | | | Reference | | | | Reference | | | |

Continued

**Table 2** Continued

| Quintiles (proportion of land use) | West Males Coefficient | LI 95% CI | UI 95% CI | P value | West Females Coefficient | LI 95% CI | UI 95% CI | P value | East Males Coefficient | LI 95% CI | UI 95% CI | P value | East Females Coefficient | LI 95% CI | UI 95% CI | P value |
|---|---|---|---|---|---|---|---|---|---|---|---|---|---|---|---|---|
| 2 | −5.04 | −12.40 | 2.32 | 0.16 | −9.40 | −19.77 | 0.97 | 0.07 | −5.55 | −19.15 | 8.04 | 0.38 | −7.79 | −18.77 | 3.19 | 0.15 |
| 3 | −6.42 | −10.46 | −2.39 | 0.00 | −10.60 | −18.98 | −2.23 | 0.02 | −2.28 | −15.09 | 10.53 | 0.70 | −2.19 | −10.70 | 6.31 | 0.58 |
| 4 | −7.94 | −14.52 | −1.35 | 0.02 | −7.23 | −14.35 | −0.12 | 0.05 | −5.12 | −16.15 | 5.92 | 0.33 | −10.66 | −17.95 | −3.37 | 0.01 |
| 5 | −7.87 | −15.28 | −0.46 | 0.04 | −8.46 | −18.68 | 1.77 | 0.10 | −6.14 | −19.55 | 7.26 | 0.33 | −12.35 | −16.73 | −7.98 | <0.001 |
|  | (Wald test of parameter: 0.029) | | | | (Wald test of parameter: 0.145) | | | | (Wald test of parameter: 0.814) | | | | (Wald test of parameter: 0.001) | | | |
| **Residential proportion** | | | | | | | | | | | | | | | | |
| 1 | Reference | | | | Reference | | | | Reference | | | | Reference | | | |
| 2 | 1.13 | −5.96 | 8.22 | 0.74 | 0.82 | −13.05 | 14.68 | 0.90 | −0.70 | −16.68 | 15.27 | 0.93 | −6.19 | −18.52 | 6.15 | 0.29 |
| 3 | 2.77 | −2.63 | 8.17 | 0.29 | 0.23 | −9.54 | 10.00 | 0.96 | −4.58 | −16.14 | 6.97 | 0.44 | −2.68 | −14.96 | 9.60 | 0.64 |
| 4 | 3.87 | −0.53 | 8.26 | 0.08 | 3.86 | −0.96 | 8.68 | 0.11 | −2.50 | −13.04 | 8.04 | 0.64 | −5.32 | −14.18 | 3.54 | 0.21 |
| 5 (Most) | 13.19 | 7.88 | 18.49 | <0.001 | 17.39 | 9.61 | 25.18 | <0.001 | 3.56 | −13.90 | 21.02 | 0.69 | −0.70 | −12.82 | 11.42 | 0.90 |
|  | (Wald test of parameter:<0.001) | | | | (Wald test of parameter: 0.004) | | | | (Wald test of parameter: 0.469) | | | | (Wald test of parameter: 0.155) | | | |
| **Land without current use** | | | | | | | | | | | | | | | | |
| 1 (Least) | Reference | | | | Reference | | | | Reference | | | | Reference | | | |
| 2 | 0.65 | −4.57 | 5.88 | 0.79 | −4.16 | −14.34 | 6.03 | 0.39 | 0.14 | −6.55 | 6.84 | 0.96 | −2.02 | −7.45 | 3.41 | 0.43 |
| 3 | 1.09 | −4.36 | 6.55 | 0.67 | −4.20 | −12.08 | 3.69 | 0.27 | 8.29 | −2.13 | 18.71 | 0.11 | 11.30 | 3.43 | 19.18 | 0.01 |
| 4 | 0.98 | −3.80 | 5.77 | 0.67 | −1.62 | −5.91 | 2.67 | 0.43 | 14.75 | 5.70 | 23.81 | 0.01 | 12.81 | 6.49 | 19.14 | 0.00 |
| 5 (Most) | 10.18 | 0.07 | 20.30 | 0.05 | 4.58 | −6.12 | 15.28 | 0.37 | 13.67 | 1.05 | 26.29 | 0.04 | 9.30 | 3.09 | 15.51 | 0.01 |
|  | (Wald test of parameter: 0.249) | | | | (Wald test of parameter: 0.316) | | | | (Wald test of parameter: <0.001) | | | | (Wald test of parameter: <0.001) | | | |
| **Discontinuous low-density urban fabric** | | | | | | | | | | | | | | | | |
| 1 (Least) | Reference | | | | Reference | | | | Reference | | | | Reference | | | |
| 2 | 0.46 | −6.48 | 7.40 | 0.89 | −3.85 | −13.57 | 5.87 | 0.41 | 0.67 | −14.40 | 15.73 | 0.92 | 0.19 | −18.09 | 18.47 | 0.98 |
| 3 | −0.40 | −6.95 | 6.16 | 0.90 | −5.65 | −17.07 | 5.76 | 0.30 | 2.83 | −9.82 | 15.48 | 0.63 | 1.37 | −12.03 | 14.77 | 0.82 |
| 4 | −0.58 | −12.46 | 11.31 | 0.92 | −6.61 | −27.00 | 13.79 | 0.50 | 5.38 | −8.08 | 18.84 | 0.39 | 0.88 | −17.35 | 19.10 | 0.92 |
| 5 (Most) | 7.98 | 2.22 | 13.75 | 0.01 | 7.92 | −1.62 | 17.46 | 0.10 | 1.47 | −13.93 | 16.87 | 0.84 | −3.82 | −24.54 | 16.89 | 0.69 |
|  | (Wald test of parameter: 0.005) | | | | (Wald test of parameter: 0.027) | | | | (Wald test of parameter: 0.822) | | | | (Wald test of parameter: <0.001) | | | |
| **Sports and leisure facilities** | | | | | | | | | | | | | | | | |
| 1 (Least) | Reference | | | | Reference | | | | Reference | | | | Reference | | | |
| 2 | 1.68 | −6.25 | 9.61 | 0.65 | −4.11 | −20.19 | 11.97 | 0.59 | −7.42 | −18.66 | 3.82 | 0.17 | −7.20 | −16.97 | 2.58 | 0.13 |
| 3 | 2.48 | −3.15 | 8.12 | 0.36 | 0.01 | −10.95 | 10.97 | 1.00 | −18.90 | −27.75 | −10.06 | <0.001 | −15.15 | −29.78 | −0.53 | 0.04 |
| 4 | 4.60 | −0.11 | 9.31 | 0.06 | 9.67 | 1.18 | 18.16 | 0.03 | −13.80 | −25.27 | −2.33 | 0.020 | −13.98 | −28.92 | 0.97 | 0.06 |

Continued

**Table 2** Continued

| Quintiles (proportion of land use) | West | | | | | | | | East | | | | | | | |
| --- | --- | --- | --- | --- | --- | --- | --- | --- | --- | --- | --- | --- | --- | --- | --- | --- |
| | Males | | | | Females | | | | Males | | | | Females | | | |
| | Coefficient | LI 95% CI | UI 95% CI | P value | Coefficient | LI 95% CI | UI 95% CI | P value | Coefficient | LI 95% CI | UI 95% CI | P value | Coefficient | LI 95% CI | UI 95% CI | P value |
| 5 (Most) | 10.39 | 1.82 | 18.95 | 0.02 | 14.12 | 2.88 | 25.35 | 0.02 | −19.87 | −31.05 | −8.68 | <0.001 | −15.16 | −31.38 | 1.07 | 0.06 |
| | (Wald test of parameter: 0.179) | | | | (Wald test of parameter: 0.086) | | | | (Wald test of parameter: 0.002) | | | | (Wald test of parameter: 0.259) | | | |

in cities containing the most (quintile 5). The association was not consistent or significant for men living in Eastern cities but was for women, although the trend fluctuated and only cities with greatest proportions of this land cover/use were associated with a higher SMR. A positive association between *green urban areas* and SMR was seen for Western women, and both sexes in Eastern cities (table 2, figure 3). The trend in Eastern cities suggested a threshold effect that may be substantially more important for health, with modest reductions in SMR as proportions increase from quintiles 1 to 4, and then a sudden and steep adverse association at quintile 5.

More derelict land (*land without current use*) was associated with higher SMRs in the Eastern cities, with the greatest adverse association quintiles 4 and 5. Western European cities with higher proportions of land classified as *residential* and *discontinuous low-density urban fabric* had higher SMRs, particularly for cities in quintile 5 (table 2), suggesting a critical level at which these land covers/uses become substantially more important to health.

### DISCUSSION

This is the first pan-European study to examine whether, and which, land covers/uses within a whole city are associated with mortality rates. By integrating health and environmental data from a large number of cities across Western and Eastern Europe, we found that the proportion of some specific land covers/uses within a city was related to mortality; most land covers/uses were not clearly associated with mortality and there were differences by macroregion and sex in association. We found no evidence that the overall distribution and balance of multiple land covers/uses, measured by landscape metrics, was related to mortality rates.

Overall, we observed that higher proportions of natural spaces and less dense or non-residential land cover/use was associated with lower mortality. For example, greater proportions of lower density settlement (such as *isolated structures*) showed some association with lower SMR, while SMRs were higher for Western European cities with a greater residential proportion and low-density discontinuous cover (which is principally housing). Specifically, our most consistent finding was that presence of more 'relatively wild' green spaces, such as *agricultural, semi-natural areas and wetlands* and *forests* was associated with lower SMRs, an association observed across sexes and macroregions and relatively strongly. The health benefits of contact with nature are very well established from both experimental and observational studies.[28] These kinds of spaces may be particularly restorative or conducive of leisure and recreation. Studies have shown the protective effects of natural environments and association with reduced risk of mortality. For example, two large Canadian studies examined green space using normalised difference vegetation index (NDVI) around residents' home and showed a higher NDVI was associated with lower mortality rates.[29 30] Our results for the more natural

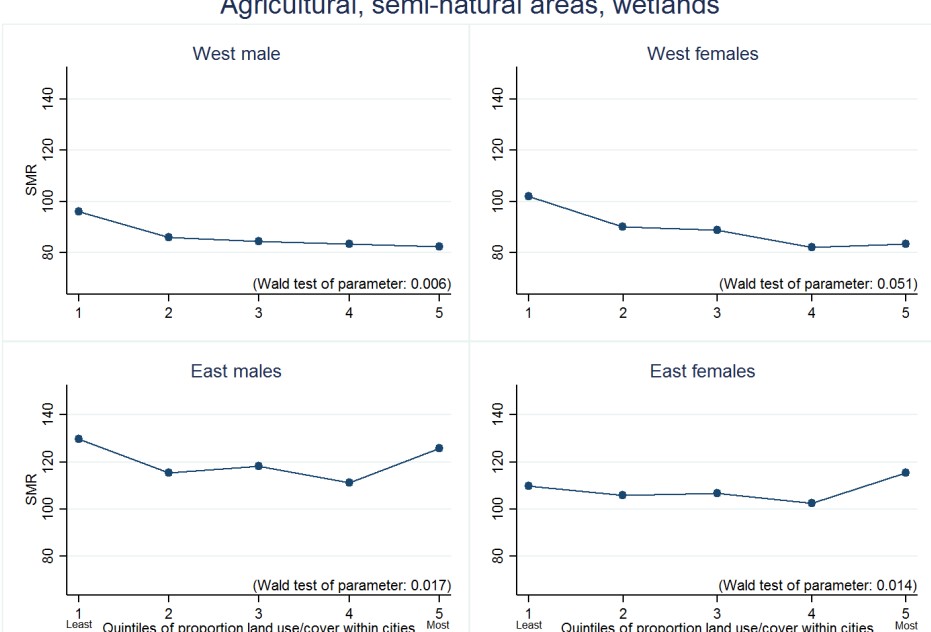

**Figure 2** Estimated standardised mortality ratios (SMRs) for cities with varying quintiles of agricultural, semi-natural areas, wetlands (quintile 1=lowest proportion).

spaces are in stark contrast to those for *green urban areas*, the proportion of which was relatively consistently associated with *higher* mortality. At first, this seems like a directly contradictory finding, but it is important to be clear what kinds of spaces are classified as 'green urban areas' in the Urban Atlas. Large urban parks are not classified solely as 'green urban areas' and will be separated into their designated components, for example, if a park contains a botanical garden, zoo or national trust gardens, these areas will be classified as an industrial or commercial areas; blue spaces within parks will be defined as waters; any space which can be used for sport or containing sport facilities (goal posts, cricket or basketball markings) will be defined as sports or leisure facilities. In this land use classification, 'green urban areas' are relatively small and manicured 'in-fill' green spaces within residential and commercial developments. Their association with greater mortality is not unprecedented in the literature. Previous work found that the total amount of green space within American cities was associated with higher mortality rates, suggesting that the indicator was capturing sprawling cities, however this study was unable to distinguish between the green space type.[31] In light of our findings, the American study may have been capturing the development of semi-natural areas into 'in-fill' green developments. We have previously found that greater amounts of '*green urban areas*' in European cities were associated with reduced reported life satisfaction for its city residents.[15] The salutogenic effects of nature may be maximised by more natural areas and/or the 'pockets of green' in and among urban developments may be insufficient to offset

the adversity of living in dense residential and commercial environments.

There were some differences between region and sex in the direction of associations. We found that less desirable land types/uses within cities, such as *land without use*, were associated with higher SMRs for those living in Eastern European cities. Cities depend on their residents for economic, social, cultural and environmental prosperity and maintaining a diverse, skilled and satisfied residential population is vital for a city since their disenchantment could trigger a vicious downward spiral.[32] Dereliction of cities has been linked to decreasing employment rates due to many individuals, particularly younger and educated, migrating from these areas to more prosperous cities[33] and individuals living in areas with a high proportion of brownfield land are significantly more likely to suffer from poorer health than those with a lower proportion.[34] Although derelict and vacant land covers/uses remain in both Western and Eastern European regions and was a national problem in many Western cities during the mid/late 20th century as many industries declined or closed,[33] there is country-specific evidence that areas of these land covers/uses are decreasing. For example, in Scotland from 2017 to 2018 derelict land decreased by 6% nationally,[35] which may reduce the importance for population health. Perhaps the approaches taken by some Western governments[36 37] to tackle it and spearhead regeneration in these areas, for example, have mitigated impacts. Furthermore, the impact of *industrial, commercial, public and military* spaces within a city seemed benign for men living in Eastern European Cities but not

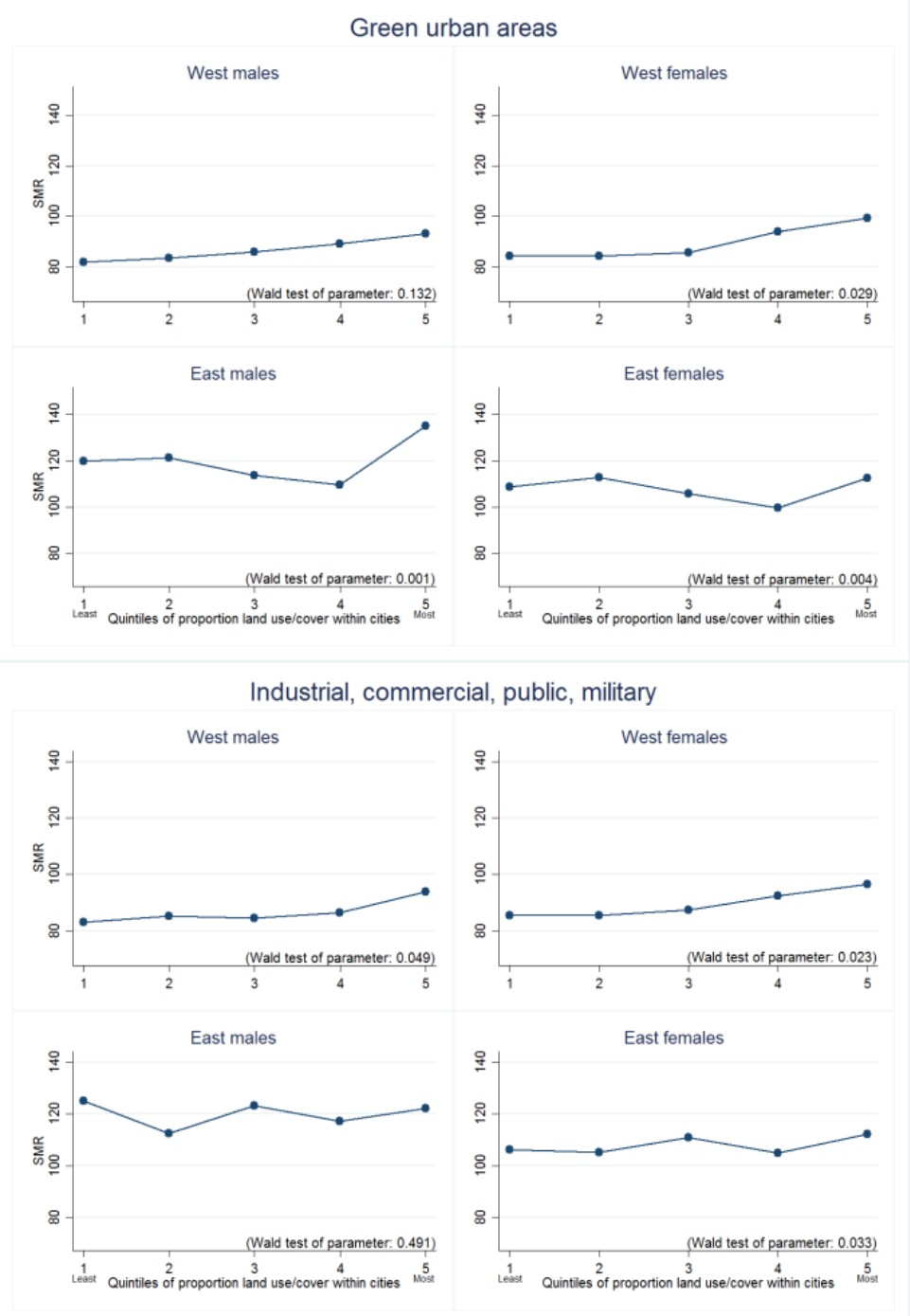

**Figure 3** Comparisons between standardised mortality ratio (SMR) and quintiles land covers/uses for green urban areas and industrial, commercial, public, military (partially universal importance by macroregion and/or sex).

for women, and both sexes in Western European cities. Males and females in eastern cities also seemed to benefit from greater proportions of *sports and leisure facilities*. The western/eastern differences may reflect levels of city and economic development not adequately captured by our control for GDP. Perhaps, for example, a city in the east that has substantial amounts of land dedicated to *sports and leisure facilities* hosts a population with sufficient affluence to sustain them.

Overall, our results present a challenge to healthy urban planning. Building on natural green spaces can address housing shortages, increase the local taxation base and support the development of local infrastructure (school, transportation, etc). Residential settlements with 'green views' command a premium price. Furthermore, a central message of contemporary urban planning is that dense and/or compact cities are 'sustainable'. Yet, the literature already hints that compact cities

characterised by high-density residential areas have both benefits and disadvantages for their residents. The environmental benefits, owing to the reduced carbon emissions required in intraurban transport and service, have been well described,[4] and yet at the same time result in a reduction in quality of life measures.[38] Our findings add to this debate by suggesting that retaining more wild and unstructured green space within cities is important for health.

### Strengths and weaknesses

This was a large international study of 233 cities within 24 countries. We were able to use high-quality, internationally comparable land cover/use data from the European Urban Atlas for each city to assess the configuration and quantity of these within each city. By including a large number of cities and countries with differing economic position, we believe our results can be generalised to similar cities globally. We were able to dissect the impact of land covers/uses both collectively (via landscape metrics) and individually. We surpass some previous research by including a very large range of land covers/uses, rather than studying the impact of just one or two.[39] We were able to match previously validated SMR data to land cover/use data for the same areas and a close time point. SMRs are robust, valid and widely used measures of population health. It is important to consider that land covers/uses within cities will co-occur, for example, dense compact cities may have a high proportion of both industrial and high-density residential land covers/uses. A strength of our analysis was that we assessed each land cover/use separately, allowing us to examine whether if similar types of land covers/uses were important for health. The aim of our study was not to understand the influence or configuration of combinations of land cover/use on health but nonetheless this offers an important line of enquiry for future research.

Although we described associations between land covers/uses and SMRs, we have no information on how the individuals used or were exposed to the land covers/uses that we included in our analysis, for example, if they live in a part of the city without that particular land use, and were unable to distinguish between land covers and land uses. This may be most pertinent for the most socioeconomically deprived individuals who may have a smaller activity space and therefore the affordances of the city-wide environment on their mortality may be limited. Data were aggregated to the city level and therefore may be subjected to both the ecological fallacy and modifiable areal unit problem. The development of the sample cities may also have changed considerably during individuals' life courses and the individual may not have always resided in that city, thus impacting on SMRs. As this was an ecological study and only measured land cover/use at one point in time and therefore unable to determine causality. We explored the possible effects of cultural and economic differences in land covers/uses by analysing our cities by macroregion, however we did not explore the different cultural meanings for each city to be able to ascertain if culturally affirming landscapes were health beneficial. The mortality indicator we used for the study was a city-level population-level measure and therefore we were unable to explore inequality in mortality by socioeconomic status.

### CONCLUSION

We found that the proportion of specific land covers/uses within cities is associated with mortality rates there, but that these associations varied by macroregion and by sex. There is great pressure to build on wild green space for economic gain and sustainability, and to promote compact cities as sustainable. Our study suggests that urban planners and developers need to regard retaining more wild and unstructured green space as important for healthy city systems, and that they should reconsider the push for dense and compact cities.

**Contributors** JRO, NN and RM contributed to the conception and design of the study. NN performed the statistical analysis. JRO performed the geospatial analysis. RM, GM, JP and NS were involved in previous work developing the main outcome measure. JRO prepared the first draft of the manuscript, with all authors contributing to its main content and revising it with critical comments. All authors have read and approved the manuscript prior to submission, and agree to be accountable for all aspects of the work in ensuring that questions related to the accuracy or integrity of any part of the work are appropriately investigated and resolved.

**Funding** JO, NN and RM are employed by the University of Glasgow and funded as part of the Neighbourhoods and Communities Programme (MC_UU_12017/10) (SPHSU10) at the MRC/CSO Social and Public Health Sciences Unit (SPHSU).

**Map disclaimer** The depiction of boundaries on this map does not imply the expression of any opinion whatsoever on the part of BMJ (or any member of its group) concerning the legal status of any country, territory, jurisdiction or area or of its authorities. This map is provided without any warranty of any kind, either express or implied.

**Competing interests** None declared.

**Patient consent for publication** Not required.

**Provenance and peer review** Not commissioned; externally peer reviewed.

**Data availability statement** Standardised Mortality Ratio data are available on reasonable request by emailing richard.mitchell@glasgow.ac.uk. European Urban Atlas data may be obtained freely from a third party, the European Environment Agency (https://www.eea.europa.eu/data-and-maps/data/urban-atlas), provided by the European Commission of the European Union.

**ORCID iD**
Jonathan R Olsen http://orcid.org/0000-0002-5356-8615

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
