## [Reviewer comments · BMJ Open]

ARTICLE DETAILS

TITLE (PROVISIONAL)	Which urban land covers/uses are associated with residents' mortality? A cross-sectional, ecological, pan-European study of 233 cities.
AUTHORS	Olsen, Jonathan; Nicholls, Natalie; Moon, Graham; Pearce, Jamie; Shortt, Niamh; Mitchell, Richard

VERSION 1 - REVIEW

REVIEWER	Meghann Mears University of Sheffield, UK
REVIEW RETURNED	09-Sep-2019

GENERAL COMMENTS	This paper addresses an important and relevant question about how urban greenspace affects population health. I congratulate the authors on their work to make large datasets on urban land cover/use and mortality rates tractable for analysis. A particular strength is the number and diversity of cities included, making the results more generalisable than those from many similar studies. However, I feel that at present there are some methodological issues that I would like to see either addressed or explained, in order to improve the robustness of the statistical results. 1. Page 5 lines 7-8, "The Atlas distinguishes 26 different land cover/use categories at a 10m² resolution". The document you reference for this also states that the minimum mappable unit is 0.25ha. Given that the most vulnerable people in a population are often limited to the areas closest to home (e.g. see references below), does this impact on the ability of your data to represent the affordances that a city provides to individuals with limited mobility and who are, in some cases, likely to be at higher risk of mortality? Talen E (2003) Neighborhoods as service providers: A methodology for evaluating pedestrian access. Environment and Planning B: Planning and Design 30(2): 181–200. DOI: 10.1068/b12977. Maas J, Verheij RA, de Vries S, et al. (2009) Morbidity is related to a green living environment. Journal of Epidemiology & Community Health 63(12): 967–973. DOI: 10.1136/jech.2008.079038. 2. Page 5 lines 16-19, "We calculated Shannon's diversity index (SDI) and Shannon's Evenness index (SEI) to assess the variety of all land covers/uses, and their relative availability, within each city". Please could you give more details of the calculation of the evenness index and the rationale for your decision – did you use 26 (i.e. all possible land covers/uses) or the total number of land covers/uses within each city as the denominator? In my view the
--

former makes the index directly comparable between areas, but it also means that the diversity and evenness indices are exactly correlated, making them redundant.

3. Page 5 lines 25-26, "Mortality was measured using all-cause standardised mortality ratios (SMR) for males and females separately, calculated for 233 European cities by Richardson et al". It isn't immediately obvious (without following the reference) that the ratios are age-group standardised. It might be helpful to make this explicit.

4. Page 6 lines 16-17, "To do this, the proportion of the total area covered by each land measure was calculated within each city, and then classified into quintiles". Representing continuous data as categorical has a number of disadvantages in regression (see reference below), including loss of statistical power and difficulty in cross-study comparison (or indeed comparison of your Eastern and Western macrozones, because you have calculated the quantile boundaries separately for these). I would recommend an alternative approach e.g. transformation of the variable or use of polynomial terms unless there is a reason why this is not possible. Bennette C & Vickers A (2012). Against quantiles: categorization of continuous variables in epidemiologic research, and its discontents. BMC Medical Research Methodology 12: 21. DOI: 10.1186/1471-2288-12-21.

5. Page 6 lines 18-21, "Each land measure was assessed separately ... All analyses were run separately by sex and macroregion". Undertaking separate analyses for each land cover ignores any possible collinearity between the different land covers/uses (e.g. land cover 1 may only be significantly associated with mortality because of its correlation with land cover 2). Undertaking separately analyses for each sex and macroregion, instead of including these as interaction terms, has reduced your sample size per analysis (and thereby reduced your ability to detect significant effects) and is also a less statistically robust way of detecting differences between these groups. I feel that including more of the predictors, plus interaction terms, in your regression models simultaneously would yield more robust results, while post-hoc investigations would enable probing of the individual significant terms to aid interpretation. Related to this, I question whether the marginal means plots are truly marginal means plots, when the other predictor terms are not accounted for.

6. Page 7 line 6: "morality" should read "mortality".

7. Page 9-10 Table 2: would be easier to read if there was something to draw the eye to significant results e.g. bold font or star notation.

8. Page 11 lines 14-28, "The two land cover/uses most consistently associated ... particularly for cities in quintile 5 (Table 2)". Several times in this paragraph you mention unexpectedly large changes in group means for either the highest or lower quintiles. Is this evidence of critical levels at which the amount of particular land covers/uses become substantially more important to health?

9. Discussion, pages 12-14. You seem to be saying that sprawling cities – a term which to me suggests urban sprawl, i.e. larger,

lower density residential lots – are associated with higher mortality rates, due to the significant association between higher levels of green urban areas and mortality. However, a more direct indicator of urban sprawl might be the amount of low density continuous urban fabric, so again I think there may be an issue of spurious results due to confounding. Moreover, you also recommend against dense and compact cities, i.e. cities with a low level of urban sprawl due to high density residential lots. These comments are contradictory. Is there a possibility of confounding with city size/area (which would be resolved by using proportions rather than areas) or population?

10. Discussion, pages 12-14. You do not mention the possible effects of different land covers/uses having different cultural meanings and/or marginal values in different areas. E.g. in Scandinavia, forests are abundant within city administrative boundaries, whereas this is not the case for example in Italy. A culturally affirming landscape is more likely to have health benefits. Related to this, given that certain land covers are naturally more common in certain biomes, there is a possibility of confounding with other characteristics of areas on a general (though messy) north-south gradient but which aren't directly climate-related.

11. Discussion, pages 12-14. Looking at the European Urban Atlas documentation, I would disagree with your statement that “green urban areas” are largely “in-fill” green spaces. This category includes types of greenspace that can be quite large – and given the minimum mappable unit of the EUA, small genuinely in-fill greenspaces are unlikely to be captured. I would like to see discussion of why you think that parks, zoos, public gardens etc. are associated with higher mortality on a between-city scale (without resort to the suggestion of urban sprawl – see above comment), when studies within cities frequently find the opposite pattern. Related to this, is there a possibility that the opposite relationships with “green urban areas” and agriculture/semi-natural/wetland are due simply to the latter being substantially larger?

12. Discussion, pages 12-14. I would like to see additional discussion of the policy implications of your study. You have stated that it is already well-documented that high-density cities are more ecologically sustainable but have negative consequences for population health. What do your findings add to this?

13. Page 12 lines 11-12, “For example, greater proportions of lower density settlement (such as isolated structures) showed some association with lower SMR”. According to supplementary table 1, this category represents isolated residential structures such as farm houses. This land use seems likely to be confounded with agricultural land cover, which you have also found to be significant in the same direction. I would like to see a model that includes both of these terms simultaneously to ensure that this association is not spurious (see also comment above).

14. Page 13 lines 12-14, “Dereliction of cities has been linked to decreasing employment rates due to many individuals, particularly younger and educated, migrating from these areas to more prosperous cities”. I'm not sure I follow this argument. Why would people leaving cities lead to lower employment rates?

	15. Page 13 lines 16-17, “Dereliction was a national problem in many Western cities during the mid/late 20th century as many industries declined or closed, but may now be less important for population health”. Less important relative to what? I would like to see a citation for this – my experience of UK data is that deprivation is still high in most ex-industrial northern cities, so I would expect population health to be worse too. 16. Page 13 lines 21-22, “Males in eastern cities also seemed to benefit from greater proportions of sports and leisure facilities”. Table 1 says that both sexes benefit from this. 17. Strengths and weaknesses, page 14. As well as the issues that I have highlighted in the above comments, additional limitations include:  - Causation cannot be determined from an ecological study. This is particularly important given the high risk of residual confounding and reverse causation, for reasons I have stated in the above comments. - Data at aggregated units e.g. cities may be subject to both the ecological fallacy and the modifiable areal unit problem. - Due to the data you have not been able to differentiate between land covers and land uses.
--	--

REVIEWER	Ebru Ersoy Aydın Adnan Menderes University
REVIEW RETURNED	19-Sep-2019

GENERAL COMMENTS	Thank you for the opportunity to review this manuscript. The technical, structural and formal quality of the manuscript is good. The paper is well written and referenced. I enjoyed reading this paper. It is a useful contribution to the literature on the relationship between land covers/uses with residents’ mortality at the international level. The paper used an innovative approach that takes into account a wide range of land covers/uses, rather than studying the impact of just one or two types *which provides a more complete picture of associations between healthy city systems and residents. Looking forward to seeing this paper published.
---

VERSION 1 – AUTHOR RESPONSE

Reviewer: 1

This paper addresses an important and relevant question about how urban greenspace affects population health. I congratulate the authors on their work to make large datasets on urban land cover/use and mortality rates tractable for analysis. A particular strength is the number and diversity of cities included, making the results more generalisable than those from many similar studies. However, I feel that at present there are some methodological issues that I would like to see either addressed or explained, in order to improve the robustness of the statistical results.

1. Page 5 lines 7-8, "The Atlas distinguishes 26 different land cover/use categories at a 10m2 resolution". The document you reference for this also states that the minimum mappable unit is 0.25ha. Given that the most vulnerable people in a population are often limited to the areas closest to home (e.g. see references below), does this impact on the ability of your data to represent the affordances that a city provides to individuals with limited mobility and who are, in some cases, likely to be at higher risk of mortality?

Talen E (2003) Neighborhoods as service providers: A methodology for evaluating pedestrian access. *Environment and Planning B: Planning and Design* 30(2): 181–200. DOI: 10.1068/b12977.

Maas J, Verheij RA, de Vries S, et al. (2009) Morbidity is related to a green living environment. *Journal of Epidemiology & Community Health* 63(12): 967–973. DOI: 10.1136/jech.2008.079038.

Response: The reviewer has highlighted an important. We note in the limitations section that we have no information on how the individuals used, or were exposed to, the land covers/uses that we included in our analysis (e.g. if they live in a part of the city without that particular land use). However, we did not explicitly state that the most socioeconomically deprived may have a smaller activity space and therefore the affordances of the city wide environment on their mortality may be limited. We have now added this to the Discussion (Section: 'Strengths and weaknesses', Page 17, lines 17 to 22):

"However, although we described associations between land covers/uses and SMRs, we have no information on how the individuals used or were exposed to the land covers/uses that we included in our analysis, for example if they live in a part of the city without that particular land use. This may be most pertinent for the most socioeconomically deprived individuals who may have a smaller activity space and therefore the affordances of the city wide environment on their mortality may be limited"

2. Page 5 lines 16-19, "We calculated Shannon's diversity index (SDI) and Shannon's Evenness index (SEI) to assess the variety of all land covers/uses, and their relative availability, within each city". Please could you give more details of the calculation of the evenness index and the rationale for your decision – did you use 26 (i.e. all possible land covers/uses) or the total number of land covers/uses within each city as the denominator? In my view the former makes the index directly comparable between areas, but it also means that the diversity and evenness indices are exactly correlated, making them redundant.

Response: The SDI and SEI was calculated only for land uses that were within that city. This provided a relative measure of the diversity of land covers/uses available at the city level (SDI) and the distribution of area among the land uses present within a city (SEI). We agree that including all land uses would have made the SDI and SEI less useful and our analysis does not directly compare the same land uses for all cities. The SDI measure we used calculated the index as: 'The index will equal zero when there is only one patch in the landscape and increases as the number of patch types or proportional distribution of patch types increases.'

For SEI: 'the index is equal to zero when the observed patch distribution is low and approaches one when the distribution of patch types becomes more even'.

This allows the SDI and SEI to be comparable but not dependent on having the same land uses/covers within each city.

We have now clarified this within the text (Section: 'Methods: Geoprocessing of Urban Atlas data', Page 6, lines 20 to 22):

"We calculated Shannon's diversity index (SDI) and Shannon's Evenness index (SEI) to assess the variety of all land covers/uses, and their relative availability, within each city. Only land covers/uses present within each city were included, this provided a relative measure of the diversity of land cover/uses available at the city level (SDI) and the distribution of area among the land cover/uses present within a city (SEI)."

3. Page 5 lines 25-26, "Mortality was measured using all-cause standardised mortality ratios (SMR) for males and females separately, calculated for 233 European cities by Richardson et al". It isn't immediately obvious (without following the reference) that the ratios are age-group standardised. It might be helpful to make this explicit.

Response: We have updated the text to state this clearly (Section: 'Methods: European mortality data (Standardised Mortality Ratios)', Page 6, line 28):

"Mortality was measured using all-cause and age-group standardised mortality ratios (SMR) for males and females separately, calculated for 233 European cities by Richardson et al."

4. Page 6 lines 16-17, "To do this, the proportion of the total area covered by each land measure was calculated within each city, and then classified into quintiles". Representing continuous data as categorical has a number of disadvantages in regression (see reference below), including loss of statistical power and difficulty in cross-study comparison (or indeed comparison of your Eastern and Western macrozones, because you have calculated the quantile boundaries separately for these). I would recommend an alternative approach e.g. transformation of the variable or use of polynomial terms unless there is a reason why this is not possible.

Bennette C & Vickers A (2012). Against quantiles: categorization of continuous variables in epidemiologic research, and its discontents. *BMC Medical Research Methodology* 12: 21. DOI: 10.1186/1471-2288-12-21.

Response: The reviewer has highlighted an important point about the dataset and as the land cover/uses were not linear we could not treat this as a continuous outcome. Initially, the modelling was attempted using city land cover sizes in m², controlling for overall city size. However, the disparity in city sizes results in extremely poor models, and convergence issues in some cases, so it was decided to standardise the measures, by conversion to percentage of the area. While we agree that use of polynomials would preserve data, for more practical interpretation and possible future guidance by public bodies, we chose to use ranking and quintiles. This also allowed us to generate the marginal estimates, which were plotted. It should be noted that use of categorisation is not without merit, and is more practical for the meaningful interpretation by policy makers, as highlighted by Barrio, I., Arostegui, I., Rodríguez-Álvarez, M. X., & Quintana, J. M. (2017). A new approach to

categorising continuous variables in prediction models: Proposal and validation. *Statistical methods in medical research*, 26(6), 2586-2602.

We have added the additional text to the manuscript (Section: Statistical Analysis, Page 7, lines 21-23):

“We categorised the data into quintiles rather than apply polynomial terms in order to aid interpretation and therefore help guide policymakers. Further, this approach also allowed us to generate the marginal estimates, which were plotted.”

5. Page 6 lines 18-21, “Each land measure was assessed separately ... All analyses were run separately by sex and macroregion”. Undertaking separate analyses for each land cover ignores any possible collinearity between the different land covers/uses (e.g. land cover 1 may only be significantly associated with mortality because of its correlation with land cover 2).

Undertaking separately analyses for each sex and macroregion, instead of including these as interaction terms, has reduced your sample size per analysis (and thereby reduced your ability to detect significant effects) and is also a less statistically robust way of detecting differences between these groups. I feel that including more of the predictors, plus interaction terms, in your regression models simultaneously would yield more robust results, while post-hoc investigations would enable probing of the individual significant terms to aid interpretation. Related to this, I question whether the marginal means plots are truly marginal means plots, when the other predictor terms are not accounted for.

Response: We understand the concern of the reviewer here, however if there were relationships between many land cover/uses and mortality this would have resulted in a substantial amount of significant findings, not just a few as we found here. We were also concerned about possible collinearity from the outset, choosing to run each land cover/use separately due to the likelihood of multicollinearity, which was confirmed when tested (Eastern European Region – VIF: 17.20, Western European Cities – VIF: 6.41). We have included this text within the manuscript and appropriate references (listed below*).

The analyses were performed separately for macroregion and gender as previous studies of mortality rates across these cities have shown that the distributions in outcome (SMR) being different. This strategy was outlined within the introduction of the manuscript (Section: Introduction, Page 5, lines 10 to 14) referencing previous research of SMR rates in Europe (Ref: Richardson, Elizabeth A., et al. "Multi-scalar influences on mortality change over time in 274 European cities." *Social Science & Medicine* 179 (2017): 45-51.) The substantially different VIF values by macroregion also indicate that pooling eastern and western cities would be inappropriate.

Existing text:

“Each land measure was assessed separately using Wald statistics and subjected to post-hoc pairwise comparisons using Sidak’s correction for multiple testing across all terms[22]. We chose not to run models ”

Updated text (Section: Statistical Analysis, Page 7 lines 25-27 and Page 8, lines 1*2):

“Each land measure was assessed separately using Wald statistics and subjected to post-hoc pairwise comparisons using Sidak’s correction for multiple testing across all terms[22]. We chose not to include all land cover/uses in a singular model due to concern of multicollinearity, which was confirmed when tested (Eastern European Region – VIF: 17.20, Western European Cities – VIF: 6.41).”

*Daoud, Jamal I. "Multicollinearity and regression analysis." *Journal of Physics: Conference Series*. Vol. 949. No. 1. IOP Publishing, 2017.

Akinwande, Michael Olusegun, Hussaini Garba Dikko, and Agboola Samson. "Variance inflation factor: as a condition for the inclusion of suppressor variable (s) in regression analysis." *Open Journal of Statistics* 5.07 (2015): 754.

6. Page 7 line 6: “morality” should read “mortality”.

Response: We thank the reviewer for highlighting this error and this has now been changed.

7. Page 9-10 Table 2: would be easier to read if there was something to draw the eye to significant results e.g. bold font or star notation.

Response: Significant results have been highlighted using bold font and we have noted this in the table heading.

8. Page 11 lines 14-28, “The two land cover/uses most consistently associated ... particularly for cities in quintile 5 (Table 2)”. Several times in this paragraph you mention unexpectedly large changes in group means for either the highest or lower quintiles. Is this evidence of critical levels at which the amount of particular land covers/uses become substantially more important to health?

Response: We had suggested but not explicitly stated that having a particularly large amount of a land cover/use in the city is suggestive of evidence of a critical level at which that certain land cover/use becomes substantially more important to health. For green urban areas we suggested this that steep adverse association at quintile 5 in eastern European cities was suggestive of a threshold

effect. We have now expanded this section state (Section Results, Land cover/uses associated with higher mortality, Page 12, Lines 21 to 22):

“a threshold effect that may be substantially more important for health”.

(Section Results, Land cover/uses associated with higher mortality, Page 13, Lines 1 to 2)

“The trend in Eastern cities suggested a threshold effect, with modest reductions in SMR as proportions increase from quintiles 1 to 4, and then a sudden and steep adverse association at quintile 5, suggesting a critical level at which these land cover/uses become substantially more important to health.”

9. Discussion, pages 12-14. You seem to be saying that sprawling cities – a term which to me suggests urban sprawl, i.e. larger, lower density residential lots – are associated with higher mortality rates, due to the significant association between higher levels of green urban areas and mortality. However, a more direct indicator of urban sprawl might be the amount of low density continuous urban fabric, so again I think there may be an issue of spurious results due to confounding. Moreover, you also recommend against dense and compact cities, i.e. cities with a low level of urban sprawl due to high density residential lots. These comments are contradictory. Is there a possibility of confounding with city size/area (which would be resolved by using proportions rather than areas) or population?

Response: We agree with the reviewer’s assessment and have carefully checked the full manuscript and have removed any reference to sprawling in relation to our own findings. Where we have used the term ‘sprawling cities’, this was describing other research that was unable to distinguish between land uses/covers within a city - similar the reviewers point, we then suggest this was not in fact urban sprawl but lower density areas of the city with in-fill green spaces rather than natural or agricultural areas.

Please find below text used within the discussion below (Page 15, lines 8-12):

“Previous work found that the total amount of green space within American cities was associated with higher mortality rates, suggesting that the indicator was capturing sprawling cities, however this study was unable to distinguish between the green space type[28]. In light of our findings, the American study may have been capturing the development of semi-natural areas into ‘in-fill’ green developments.”

Please also see response to question 13, we have provided additional text describing co-occurrence of land cover/uses within cities.

10. Discussion, pages 12-14. You do not mention the possible effects of different land covers/uses having different cultural meanings and/or marginal values in different areas. E.g. in Scandinavia, forests are abundant within city administrative boundaries, whereas this is not the case for example in Italy. A culturally affirming landscape is more likely to have health benefits. Related to this, given that

certain land covers are naturally more common in certain biomes, there is a possibility of confounding with other characteristics of areas on a general (though messy) north-south gradient but which aren't directly climate-related.

Response: We have explored the possible effects of cultural and economic differences in land covers/uses by analysing our cities by Western and Eastern region, however we did not explore the cultural meanings for each city. We did carefully clip each of the cities to include only their immediate city boundaries and not the wider city region that may include a greater portion of rural lands.

We have updated the strengths and weaknesses section to highlight that we did not explore cultural meanings land cover/uses within cities (Section: Discussion, Strengths and Weaknesses, Page 17, Lines 27 to 28, Page 18, lines 1 to 2):

“We explored the possible effects of cultural and economic differences in land covers/uses by analysing our cities by macroregion, however we did not explore the different cultural meanings for each city to be able to ascertain if culturally affirming landscapes were health beneficial.”

11. Discussion, pages 12-14. Looking at the European Urban Atlas documentation, I would disagree with your statement that “green urban areas” are largely “in-fill” green spaces. This category includes types of greenspace that can be quite large – and given the minimum mappable unit of the EUA, small genuinely in-fill greenspaces are unlikely to be captured. I would like to see discussion of why you think that parks, zoos, public gardens etc. are associated with higher mortality on a between-city scale (without resort to the suggestion of urban sprawl – see above comment), when studies within cities frequently find the opposite pattern. Related to this, is there a possibility that the opposite relationships with “green urban areas” and agriculture/semi-natural/wetland are due simply to the latter being substantially larger?

Response: The Urban Atlas uses this classification where there are green areas within cities that are both ‘not natural’ and have no other use, largely referring to urban green fill. We appreciate that the Urban Atlas documentation is not highly specified and we were required to conduct a ground-truthing exercise across a number of cities to understand the classifications; this showed that if a park contains a botanical garden, zoo or national trust gardens, these areas were classified as an Industrial or Commercial area; blue spaces within parks were defined as waters; any space which can be used for sport or containing sport facilities (goal posts, cricket markings) were defined as sports or leisure facilities; and urban natural areas within large urban parks defined as agricultural, semi-natural areas, wetlands. This means that huge areas of large urban parks, such as Hyde Park in London would not be defined as green urban areas.

We have expanded the section describing this land-use definition to provide clarity (Section Discussion, Page 14, Lines 25 to 28 and Page 15, lines 1 to 4):

Original text:

“but it is important to be clear what kinds of spaces are classified as ‘green urban areas’ in the Urban Atlas. In this land use classification, they are relatively small and manicured ‘in-fill’ green spaces within residential and commercial developments.”

Updated text:

“but it is important to be clear what kinds of spaces are classified as ‘green urban areas’ in the Urban Atlas. Large urban parks are not classified solely as ‘green urban areas’ and will be separated into their designated components, for example if a park contains a botanical garden, zoo or national trust gardens, these areas will be classified as an Industrial or Commercial areas; blue spaces within parks will be defined as waters; any space which can be used for sport or containing sport facilities (goal posts, cricket or basketball markings,) will be defined as sports or leisure facilities. In this land use classification, ‘green urban areas’ are relatively small and manicured ‘in-fill’ green spaces within residential and commercial developments.”

12. Discussion, pages 12-14. I would like to see additional discussion of the policy implications of your study. You have stated that it is already well-documented that high-density cities are more ecologically sustainable but have negative consequences for population health. What do your findings add do this?

Response: We have now updated the section discussing the policy implications of our study (see below):

“Overall, our results present a challenge to healthy urban planning. Building on natural green spaces can address housing shortages, increase the local taxation base and support the development of local infrastructure (school, transportation etc.). Residential settlements with ‘green views’ command a premium price. Further, a central message of contemporary urban planning is that dense and/or compact cities are ‘sustainable’. Yet, the literature already hints that compact cities characterised by high density residential areas have both benefits and disadvantages for their residents. The environmental benefits, owing to the reduced carbon emissions required in intraurban transport and service, have been well described⁴, and yet at the same time result in a reduction in quality of life measures.”

We have also included an additional sentence to conclude this paragraph stating what our findings add to this (Section Discussion, Page 16, Lines 27 to 28):

“Our findings add to this debate by suggesting that retaining more wild and unstructured green space within cities is important for health.”

13. Page 12 lines 11-12, “For example, greater proportions of lower density settlement (such as isolated structures) showed some association with lower SMR”. According to supplementary table 1, this category represents isolated residential structures such as farm houses. This land use seems likely to be confounded with agricultural land cover, which you have also found to be significant in the same direction. I would like to see a model that includes both of these terms simultaneously to ensure that this association is not spurious (see also comment above).

Response: Each area within the urban atlas has a singular definition and can only be categorised using a one or another classification, meaning the issue of confounding is not present here.

The reviewer has described that similar land uses do co-occur within cities, for example a dense/compact city may typically have greater quantities of both industrial and high density residential areas. We have now noted this is a limitation when using city level urban land cover/use data. As stated in our response to Question 5, we performed the analysis for each land cover/use separately due to multicollinearity.

We have now included discussion of the co-occurrence of land cover/uses within cities in the discussion section and stated that in future we will conduct analysis to explore the combinations of land cover/uses within cities and health. (Section Discussion, Strengths and weaknesses Page 17, Lines 10 to 16):

“It is important to consider that land cover/uses within cities will co-occur, for example dense compact cities may have a high proportion of both industrial and high density residential land covers/uses. A strength of our analysis was that we assessed each land cover/use separately, allowing us to examine whether if similar types of land cover/uses were important for health. The aim of our study was not to understand the influence or configuration of combinations of land cover/use on health but nonetheless this offers an important line of enquiry for future research.”

14. Page 13 lines 12-14, “Dereliction of cities has been linked to decreasing employment rates due to many individuals, particularly younger and educated, migrating from these areas to more prosperous cities”. I’m not sure I follow this argument. Why would people leaving cities lead to lower employment rates?

Response: Cities depend on their residents for economic, social, cultural and environmental prosperity. Maintaining a diverse, skilled, and satisfied residential population is vital for a city since their disenchantment could trigger a vicious downward spiral (Insch, Andrea. "Branding the city as an attractive place to live." City Branding. Palgrave Macmillan, London, 2011. 8-14.). Our argument is that cities with high dereliction could contribute to this vicious spiral. However, the reviewer has highlighted this was not clear within the manuscript; we have now included the following text (Section Discussion, Page 15, Lines 19 to 22):

“Cities depend on their residents for economic, social, cultural and environmental prosperity and maintaining a diverse, skilled, and satisfied residential population is vital for a city since their disenchantment could trigger a vicious downward spiral (Insch A, 2011). Dereliction of cities has been linked to decreasing employment rates due to many individuals, particularly younger and educated, migrating from these areas to more prosperous cities....”

15. Page 13 lines 16-17, “Dereliction was a national problem in many Western cities during the mid/late 20th century as many industries declined or closed, but may now be less important for population health”. Less important relative to what? I would like to see a citation for this – my

experience of UK data is that deprivation is still high in most ex-industrial northern cities, so I would expect population health to be worse too.

Response: The reviewer makes an important point. We have rephrased this section of text to state that dereliction remains a national problem in many Western problems but that governments are proactively decreasing the proportion of these lands, for example in Scotland from 2017 to 2018 derelict land decreased by 6% (Page 14).

We have updated the text (Section Discussion, Page 15, Lines 25 to 28 and page 16 lines 1 to 2):

Original text:

“Dereliction was a national problem in many Western cities during the mid/late 20th century as many industries declined or closed, but may now be less important for population health.”

Updated text:

“Although derelict and vacant land covers/uses remain in both Western and Eastern European regions and was a national problem in many Western cities during the mid/late 20th century as many industries declined or closed, there is country specific evidence that areas of these land cover/uses are decreasing. For example in Scotland from 2017 to 2018 derelict land decreased by 6% nationally, which may reduce the importance for population health”

Reference: Scottish Government. Scottish Vacant and Derelict Land Survey: 2018 Edinburgh, Scotland.: Scottish Government; 2019 [Available from: <https://www.gov.scot/publications/scottish-vacant-derelict-land-survey-2018/> accessed 27/09/2019 2019.]

16. Page 13 lines 21-22, “Males in eastern cities also seemed to benefit from greater proportions of sports and leisure facilities”. Table 1 says that both sexes benefit from this.

Response: We have updated the text to state that both males and female benefit from greater proportions of sports and leisure facilities.

17. Strengths and weaknesses, page 14. As well as the issues that I have highlighted in the above comments, additional limitations include:

- Causation cannot be determined from an ecological study. This is particularly important given the high risk of residual confounding and reverse causation, for reasons I have stated in the above comments.

Response: We have included the following text in the discussion (Section Discussion, Strengths and weaknesses, Page 17, Lines 26 to 27):

“As this was an ecological study and only measured land cover/use at one point in time and therefore unable to determine causality.”

- Data at aggregated units e.g. cities may be subject to both the ecological fallacy and the modifiable areal unit problem.

Response: We have included the following text in the discussion (Section Discussion, Strengths and weaknesses, Page 17, Lines 22 to 24):

“Data were aggregated to the city level and therefore may be subjected to both the ecological fallacy and modifiable areal unit problem.”

- Due to the data you have not been able to differentiate between land covers and land uses.

Response: We have included the following text in the discussion (Section Discussion, Strengths and weaknesses, Page 17, Lines 17 to 20):

“However, although we described associations between land covers/uses and SMRs, we have no information on how the individuals used or were exposed to the land covers/uses that we included in our analysis, for example if they live in a part of the city without that particular land use, and were unable to distinguish between land covers and land uses”

Reviewer: 2

Thank you for the opportunity to review this manuscript.

The technical, structural and formal quality of the manuscript is good. The paper is well written and referenced. I enjoyed reading this paper. It is a useful contribution to the literature on the relationship between land covers/uses with residents' mortality at the international level. The paper used an innovative approach that takes into account a wide range of land covers/uses, rather than studying the impact of just one or two types *which provides a more complete picture of associations between healthy city systems and residents.

Looking forward to seeing this paper published.

Response: We thank the reviewer for their positive comments regarding our manuscript.

VERSION 2 – REVIEW

REVIEWER	Meghann Mears University of Sheffield, UK
REVIEW RETURNED	22-Oct-2019

GENERAL COMMENTS	I would again like to congratulate the authors on their work. The additions to the text clarify the methods and discussion well. While I do not agree with all of their statements regarding their choice of methodology (e.g. that marginal estimates cannot be calculated for continuous variables and that it was not possible to include more variables in models simultaneously), I also appreciate that alternative approaches would have made the results more difficult to interpret. Thank you for the opportunity to review this study.
--